# Causal role of the dorsolateral prefrontal cortex in modulating the balance between Pavlovian and instrumental systems in the punishment domain

**Hyeonjin Kim**[1]*, **Jihyun K. Hur**[2], **Mina Kwon**[1], **Soyeon Kim**[1], **Yoonseo Zoh**[3], **Woo-Young Ahn**[1,4]

**1** Department of Psychology, Seoul National University, Seoul, Korea, **2** Department of Psychology, Yale University, New Haven, Connecticut, United States of America, **3** Department of Psychology, Princeton University, Princeton, New Jersey, United States of America, **4** Department of Brain and Cognitive Sciences, Seoul National University, Seoul, Korea

* wahn55@snu.ac.kr

**Data Availability Statement:** All code and data for reproducing the analyses are available in the GitHub repository (https://github.com/CCS-Lab/project_tDCS_public).

## Abstract

Previous literature suggests that a balance between Pavlovian and instrumental decision-making systems is critical for optimal decision-making. Pavlovian bias (i.e., approach toward reward-predictive stimuli and avoid punishment-predictive stimuli) often contrasts with the instrumental response. Although recent neuroimaging studies have identified brain regions that may be related to Pavlovian bias, including the dorsolateral prefrontal cortex (dlPFC), it is unclear whether a causal relationship exists. Therefore, we investigated whether upregulation of the dlPFC using transcranial current direct stimulation (tDCS) would reduce Pavlovian bias. In this double-blind study, participants were assigned to the anodal or the sham group; they received stimulation over the right dlPFC for 3 successive days. On the last day, participants performed a reinforcement learning task known as the orthogonalized go/no-go task; this was used to assess each participant's degree of Pavlovian bias in reward and punishment domains. We used computational modeling and hierarchical Bayesian analysis to estimate model parameters reflecting latent cognitive processes, including Pavlovian bias, go bias, and choice randomness. Several computational models were compared; the model with separate Pavlovian bias parameters for reward and punishment domains demonstrated the best model fit. When using a behavioral index of Pavlovian bias, the anodal group showed significantly lower Pavlovian bias in the punishment domain, but not in the reward domain, compared with the sham group. In addition, computational modeling showed that Pavlovian bias parameter in the punishment domain was lower in the anodal group than in the sham group, which is consistent with the behavioral findings. The anodal group also showed a lower go bias and choice randomness, compared with the sham group. These findings suggest that anodal tDCS may lead to behavioral suppression or change in Pavlovian bias in the punishment domain, which will help to improve comprehension of the causal neural mechanism.

**Funding:** WYA was supported by the Basic Science Research Program through the National Research Foundation (NRF) of Korea funded by the Ministry of Science, ICT, & Future Planning NRF-2018R1C1B3007313 and NRF-2018R1A4A1025891 (https://www.nrf.re.kr/index); the NRF grant funded by the Korea government (MSIT) No. 2021M3E5D2A0102249311 (https://www.nrf.re.kr/index); and the Creative Pioneering Researchers Program through Seoul National University (https://www.snu.ac.kr/). The funders had no role in study design, data collection and analysis, decision to publish, or preparation of the manuscript.

**Competing interests:** The authors have declared that no competing interests exist.

## Introduction

Decision-making is governed by multiple systems, including the fundamental Pavlovian and instrumental systems. The Pavlovian system involves a pre-preprogrammed behavioral tendency known as Pavlovian bias (i.e., approaching reward-predictive stimuli and avoiding punishment-predictive stimuli) [1]. In contrast, the instrumental system involves learning the optimal response to each stimulus by evaluating its outcomes without prior preparation. Although the Pavlovian bias has several benefits, it may hamper goal-directed behavior. For example, animals (e.g., pigeons) with strong Pavlovian bias fail to learn to withhold pecking in response to stimuli predictive of food, even when they can receive food only by withholding pecking [2, 3]. Humans are also affected by Pavlovian bias in various decision-making situations, such as dieting [4, 5] or substance abuse [6]. Thus, there is a need to investigate methods to effectively overcome such bias.

The neural mechanisms that underlie Pavlovian bias are not fully understood, but some previous research has suggested that the prefrontal cortex plays a pivotal role in overcoming Pavlovian bias [7–9]. A functional magnetic resonance imaging (fMRI) study of participants who successfully employed the instrumental system during conflict with the Pavlovian system found that such individuals showed hyperactivation of the bilateral inferior frontal gyri while anticipating inhibition [9]. In addition, an electroencephalography study showed that the activation of the anterior cingulate cortex, as measured by the midfrontal theta power of the electroencephalogram signal, was associated with overcoming Pavlovian bias [7]. However, these studies failed to provide conclusive evidence for a causal neural mechanism, and the brain regions that control Pavlovian bias remained unknown.

We speculated that the dorsolateral prefrontal cortex (dlPFC) might be a key region involved in controlling Pavlovian bias. The dlPFC has been implicated in higher-level cognitive control and goal-directed actions [4, 5, 10–17]. For example, dieters showed hyperactivation of the dlPFC when they successfully selected healthy food over tasty food [4]. In addition, the dlPFC was important in individuals who valued stimuli in a context-dependent manner and performed goal-directed behavior to maximize reward [10]. Although a previous fMRI experiment studying the neural correlates of Pavlovian bias did not identify the dlPFC as a candidate region [9], the negative results are related to the imaging strategy used in the study, rather than the lack of a relationship. The imaging was focused on subcortical structures; the dlPFC regions were not assessed.

In the present study, we evaluated the presence of a causal relationship between the dlPFC and Pavlovian bias using non-invasive brain stimulation (i.e., transcranial direct current stimulation [tDCS]). Using tDCS was based on several previous studies of modulating decision-making biases. For example, the competition between the model-based and the model-free systems [18], as well as affective bias of instrumental action [19], were modulated by tDCS targeting the prefrontal cortex.

Overall, we investigated whether anodal tDCS on dlPFC would suppress the Pavlovian bias (sham-controlled); we sought to identify the causal neural mechanism underlying such bias. We applied anodal tDCS over the right dlPFC [20–22] for 3 consecutive days [23–25]. On the third day, we administered a reinforcement learning task known as the orthogonalized go/no-go task, which measured the degree of Pavlovian bias [9]. The task had four conditions; two were Pavlovian-congruent, where go was the action required to win the reward and no-go was the action required to avoid punishment; the two remaining conditions were Pavlovian-incongruent, where go was the action required to avoid punishment and no-go was the action required to win the reward. Participants were required to learn the correct action for each condition to maximize the reward and minimize punishment. We compared the degree of

Pavlovian bias across tDCS groups using the difference in behavioral accuracy between Pavlovian-congruent and Pavlovian-incongruent conditions. We also used a model parameter (i.e., Pavlovian bias parameter) estimated by computational modeling and hierarchical Bayesian analysis (HBA) as another index of Pavlovian bias. Under the punishment domain, we found significantly lower Pavlovian bias in the anodal tDCS group than in the sham group.

## Materials and methods

### Participants

We recruited 39 participants from Seoul National University in Seoul, Korea, using online and offline advertisements. The experimental protocol was approved by the Seoul National University Research Ethics Committee and all participants provided informed consent before participation. Participants were excluded if they were unwilling to participate in the study, or were not fluent in Korean; they were also excluded if they reported impaired color discrimination, psychiatric medication use, neurological or psychiatric illness, or any health conditions that would make them unsuitable for the experiments. In addition, participants were excluded if they had low-quality data such as sleep during the experiment or results that indicated an inability to understand the task. Finally, we eliminated participants with a go-to-win accuracy of < 0.1 because learning failure in the easiest go-to-win condition indicated a lack of understanding or concentration. In total, data from 17 and 14 participants in the anodal and sham sessions, respectively, were analyzed (see below for more information).

### Experimental protocol

First, we collected data regarding the participants' basic demographic information (age and sex) and psychological characteristics. We administered the Structured Clinical Interview for DSM-5 to detect mental illnesses. In addition, we evaluated the psychological characteristics of obsession-compulsion (Yale-Brown Obsessive Compulsive Scale), depression (Beck's Depression Inventory), anxiety (State-Trait Anxiety Inventory), and impulsivity (Barratt Impulsiveness Scale version 11). The participants visited the laboratory for 3 consecutive days and repeated the visits to counterbalance the tDCS polarity (six total sessions). For the first 2 days, participants received tDCS for 20 min; on the third day, participants performed an orthogonalized go/no-go task after they had received tDCS stimulation for 20 min. The daily visiting time was matched on a within-participant basis to remove the confounding effect of circadian rhythm [26, 27]. Participants were randomly assigned to receive anodal or sham stimulation on the first or second 3 days of visits. The first and second sets of visits were separated by a mean of 24 days. We found significantly better performance in the second task (see S1 Fig), suggesting a practice effect. Therefore, we analyzed behavior data only from the first task to avoid any potential confounding effects.

### tDCS stimulation

During each session, tDCS was applied for 20 min using circular sponge electrodes (size = 25 cm$^2$) and the Starstim system (Neuroelectrics, Barcelona, Spain). The target electrode was positioned on the right dlPFC (i.e., F4 according to the 10–10 International 10–20 electroencephalogram electrode system); the return electrode was positioned on the left cheek (Fig 1). The stimulation protocol was based on previous studies that used tDCS targeting dlPFC [20]. The left cheek was selected as the return position to avoid confounding cortical activation [28–31]. The stimulation included 30 s of ramp-up and ramp-down at the beginning and end of the stimulation, respectively. During the anodal session, anodal stimulation to F4 was

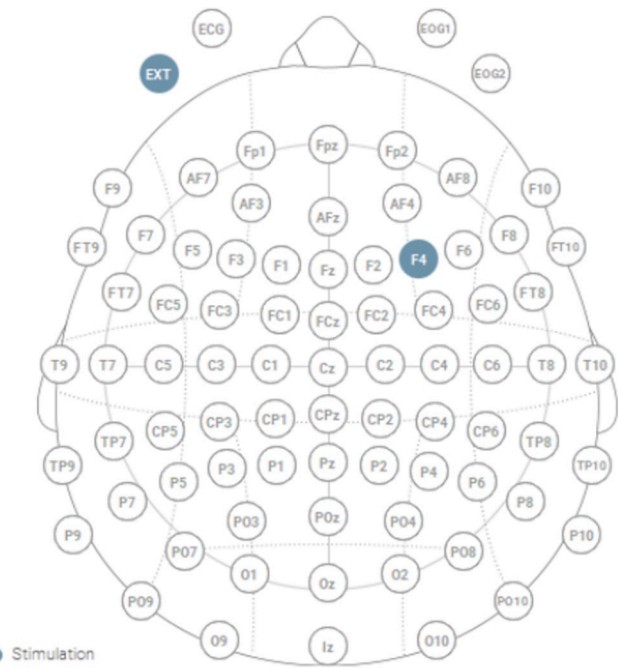

**Fig 1. Montage of tDCS.** A sponge was placed over the right dlPFC (F4) to stimulate the brain using weak electric current (2 mA) for 20 min. Another sponge was placed on the left cheek. This figure was adapted from the protocol summary panel in Starstim software NIC (copyright notice © Neuroelectrics SLU).

performed for 19 min between ramp-up and ramp-down stimulations; however, in the sham session, participants were not stimulated between the ramp-up and ramp-down stimulations. During the stimulation, participants were instructed to sit with their gaze fixed on a crosshair on the computer monitor. The double-blind mode in the Starstim software was used to ensure that all experimenters and participants remained unaware of the order of polarity. The software blinds the type of current stimulation using a 4-digit password lock set by a third-party administrator.

We employed some strategies to reduce the potential limitations of tDCS in the current stimulation protocol. First, the electrode placement and size can affect the spatial distribution of stimulation [32]. Therefore, we placed a return electrode in an extracephalic area (i.e., left chick) to minimize the stimulation of other cortical areas and the shunting effect caused by a short inter-electrode distance [33]. In addition, the effects of tDCS can be confounded by biological and lifestyle factors [33]. We mitigated such factors by stimulating participants over 3 consecutive days before the task to produce cumulative and larger effects. The participants visited the laboratory at the same time (variation of < 3 h) to reduce the effects of circadian rhythm. Finally, to reduce confounding factors related to the experimental design, we used a sham-controlled double-blind protocol.

## Experimental task

We used the orthogonalized go/no-go task reported by Guitart-Masip et al. (2012) (Fig 2). At the beginning of each trial, a 1000-ms cue (fractal image) was presented to indicate one of four conditions; go-to-win reward, go-to-avoid punishment, no-go-to-win reward, and no-go-to-avoid punishment. After a variable interval of 250–2000 ms, a target circle appeared for a maximum of 1500 ms, after which the participants responded with go or no-go within 1000 ms.

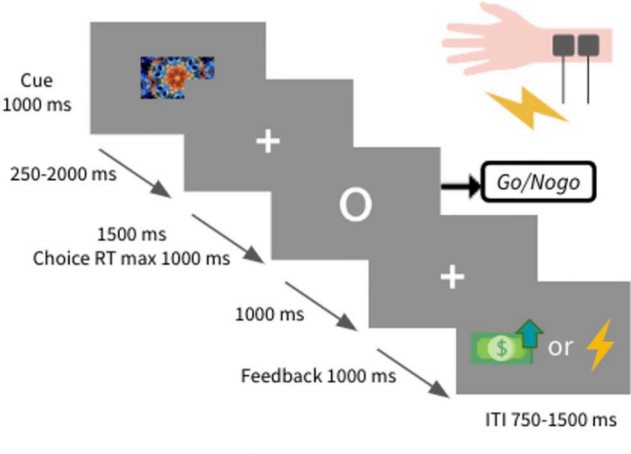

**Fig 2. Orthogonalized go/no-go task.** Four types of stimuli were presented. Two stimuli were Pavlovian-congruent: go action to win reward and no-go action to avoid punishment. The remaining two stimuli were Pavlovian-incongruent: go action to avoid punishment and no-go action to win reward. Participants were instructed to maximize reward and minimize punishment by learning the correct action for each stimulus. Participants were asked to select an action when a target was presented. The reward was a picture of money, 1000 won (approximately US$ 1), whereas the punishment was an electric shock to the wrist. RT: response time, ITI: intertrial interval.

After 1000 ms, participants received feedback according to their response and cue condition. The feedback included virtual monetary gain as a reward, a yellow bar as a neutral outcome, and an electric shock as punishment instead of monetary loss to maximize the effect of punishment [34]. The optimal response led to a beneficial outcome for each condition, with a probability of 0.7. Therefore, participants learned the optimal response to each cue from trial and error. The task included 180 trials, with 45 trials for each condition.

Although money is secondary feedback and shock is primary feedback, we decided to use monetary gain as the reward and electric shock as the punishment based on previous studies [34–37]. The electric shock was applied to each participant's left wrist. The intensity of the electric shock (2–12.4mA) was adjusted to cause a "moderately unpleasant" sensation (5 points on an 11-point Likert scale [i.e., 0 = not at all unpleasant, 10 = very un- pleasant]). (see S1 File for more information).

## Behavioral data analysis

Behavioral data were analyzed using R [38]. Response accuracy was calculated as the proportion of correct choices. The difference in accuracy between Pavlovian-congruent and Pavlovian-incongruent conditions was evaluated using Student's t-test. The behavioral Pavlovian bias index was evaluated as the difference between the accuracy of Pavlovian-congruent and

Pavlovian-incongruent conditions; it was calculated individually for each domain. For example, Pavlovian bias index in the punishment domain was calculated by subtracting the accuracy of the go-to-avoid condition from the accuracy of the no-go-to-avoid condition.

$$Pavlovian\ bias = (go\_to\_win + no\_go\_to\_avoid) - (no\_go\_to\_win + go\_to\_avoid)$$

$$Pavlovian\ bias\ (reward) = (go\_to\_win) - (no\_go\_to\_win)$$

$$Pavlovian\ bias\ (punishment) = (no\_go\_to\_avoid) - (go\_to\_avoid)$$

## Computational modeling

We tested three models. A previous study suggested that Model 1 had the best fit and consisted of five parameters (Model RW + noise + bias + Pav; Guitart-Masip et al., 2012). Model 1 calculates the probability of performing (or withholding it) an action in response to the stimulus in each trial, based on action weights. If a participant successfully learned the action-reward contingency, the probability of performing a correct action was higher. It is calculated as follows:

$$p(a_1 \mid s) = \left\{ \frac{exp(W_t(a_1, s))}{exp(W_t(a_1, s)) + exp(W_t(a_2, s))} \right\} (1 - \xi) + \frac{\xi}{2} \ldots \tag{1}$$

In particular, the go action probability was larger if the W value for go ($a_1$) was greater using squashed softmax and for no-go ($a_2$), vice versa. Here, t is the trial number ($1 \leq t \leq 180$) and s is the stimulus ($s \in \{1, 2, 3, 4\}$). The four stimuli indicate four conditions, respectively: go-to-win reward, go-to-avoid punishment, no-go-to-win reward, and no-go-to-avoid punishment. In addition, a is the action ($a \in \{0, 1\}$), where 1 is go and 0 is no-go. $\xi$ is the irreducible noise ($0 \leq \xi \leq 1$), where a value closer to 1 indicates random choice less considering the W value. W(a, s) is the action weight, which is defined as follows:

$$W_t(a, s) = \begin{cases} Q_t(a, s) + b + \pi V_t(s) & if\ a = go \\ Q_t(a, s) & otherwise \end{cases} \ldots \tag{2}$$

Q(a, s) and V(s) are updated by each trial according to the equations below:

$$Q_t(a_t, s_t) = Q_{t-1}(a_t, s_t) + \varepsilon(\rho r_t - Q_{t-1}(a_t, s_t)) \ldots \tag{3}$$

$$V_t(s_t) = V_{t-1}(s_t) + \varepsilon(\rho r_t - V_{t-1}(s_t)) \ldots \tag{4}$$

In Eq (3), r is the feedback ($r \in \{-1, 0, 1\}$), where 1 is the reward, 0 is neutral, and $-1$ is punishment. $\varepsilon$ is the learning rate ($0 \leq \varepsilon \leq 1$); If $\varepsilon$ is closer to 1, it is more likely to reflect the previous feedbacks to update Q values. Furthermore, $\rho$ is outcome sensitivity ($0 \leq \rho$). A larger $\rho$ indicates the participant subjectively exaggerates the outcome value. Using this process, the Q value converges to the high-probability outcome for each stimulus when the correct action for the stimulus is accumulated.

In Eq (4), the V value is updated in a manner similar to the Q value, but it converges to the high-probability feedback for each stimulus, regardless of the performed actions. In Eq (2), for the updated Q values when the action was go, the go bias parameter b and V value multiplied by the Pavlovian bias parameter $\pi$ ($0 \leq \pi$) were added to the Q values; they consisted of the W

values. A large go bias was correlated with large W(go, s). When the V value converged to reward, and the action was go, the large Pavlovian bias parameter was correlated with generally large W(go, reward) and generally small W(no-go, reward). When the V value converged to punishment and the action was go, the large Pavlovian bias parameter was correlated with generally small W(go, punishment) and generally large W(no-go, punishment). This suggests that a large Pavlovian bias parameter was correlated with greater predisposition to Pavlovian-congruent choices.

Model 2 shares almost all equations and updating rules with Model 1, although it has distinct feedback sensitivity parameters for reward and punishment cues: $\rho_{rew}$, and $\rho_{pun}$, respectively. Therefore, Model 2 contains six parameters. Model 3 shares almost all equations and updating rules with Model, although it has different Pavlovian bias parameters for reward and punishment cues: $\pi_{rew}$ and $\pi_{pun}$, respectively. Model 3 contains seven parameters and was used to test the distinct effect found in behavioral data, where Pavlovian bias was only significant in the punishment domain.

## Model parameter estimation using HBA

The model parameters were estimated using HBA [39–41]. HBA has some advantages over the traditional maximum likelihood estimation (MLE) method. First, HBA provides estimated parameters as posterior distributions, rather than the point estimates provided by MLE. The distributions provide additional information, particularly regarding the uncertainty of estimated values. Second, the hierarchical structure of HBA allows stable and reliable estimation of individual parameters. Individual-level MLE estimates are often noisy and unreliable; group-level MLE estimates do not include information concerning individual differences. In HBA, each individual estimate informs the group estimate (hyperparameter), and the individual commonalities reflected in the hyperparameter inform individual estimates. Therefore, individual estimates are more stable and reliable, even when data are insufficient. Previous studies have found that parameters estimated by HBA are more accurate than parameters estimated by MLE [42].

We separately fitted the models for anodal and sham groups to make stable and reliable individual estimates that reflected similarities within each group. HBA was conducted by hBayesDM (v. 1.1.1) [43] and R Stan (v. 2.21.0) [44]. Stan is a probabilistic program used for Bayesian modeling; it provides inferences based on Markov chain Monte Carlo (MCMC) algorithms, such as the Hamiltonian Monte Carlo, for sampling from high-dimensional parameter spaces. Weakly informative priors were used to reduce their influence on the posterior distributions [43]. In addition, non-centered parameterization (Matt trick) was used to optimize the sampling process [45]. We used four independent chains and a sample size of 4000, including 2000 burn-in samples per chain. The use of four independent chains ensured that the estimated parameters were stable, despite variations in the starting points [46]. We also confirmed the accuracy of parameter estimation by inspecting well-mixed trace plots and the Rhat values (Rhat < 1.1).

## Model comparison

We used LOOIC to compare the models [47]. The LOOIC value for each model was calculated by estimating the out-of-sample prediction accuracy of the fitted models. This method uses the log-likelihood from posterior simulations of the estimated parameters. We used R package loo to identify the model with the lowest LOOIC value, which had the best fit [47].

## Group comparison of model parameters

For each group-level parameter, we subtracted the posterior distribution of the sham group from the posterior distribution of the anodal group for analysis of group-level differences. Group differences were considered credible when the 95% highest density intervals of posterior difference distributions did not include the value 0 [48].

# Results

## Anodal and sham group characteristics

We analyzed data from 31 participants, including the basic demographic information (age and sex), psychiatric symptoms, and psychological characteristics (Table 1). There were no significant group-level differences in terms of demographic, psychiatric, and psychological variables

**Table 1. Descriptive statistics.**

|  | Sham (N = 14) | Anode (N = 17) | p value |
|---|---|---|---|
| **Age** | 25.071 (3.731) | 23.529 (3.484) | 0.245 |
| **Sex; male** | 5 (35.7%) | 8 (47.1%) | 0.524 |
| **SCID**[a] |  |  |  |
| avoidant | 2.357 (2.098) | 2.294 (1.724) | 0.927 |
| dependent | 1.286 (1.773) | 1.412 (1.228) | 0.817 |
| obsessive-compulsive | 2.714 (1.816) | 3.353 (1.656) | 0.315 |
| passive-aggressive | 1.000 (1.109) | 1.353 (1.967) | 0.555 |
| depressive | 1.714 (1.858) | 2.059 (1.919) | 0.618 |
| paranoid | 1.500 (1.871) | 1.706 (1.929) | 0.767 |
| schizotypal | 0.857 (1.027) | 0.765 (1.480) | 0.845 |
| schizoid | 0.929 (1.207) | 1.412 (1.502) | 0.339 |
| histrionic | 1.929 (1.269) | 2.176 (1.590) | 0.640 |
| narcissistic | 3.429 (2.503) | 3.353 (2.783) | 0.938 |
| borderline | 1.429 (1.742) | 2.941 (3.750) | 0.176 |
| antisocial | 0.857 (1.406) | 0.412 (0.870) | 0.289 |
| **Y-BOCS**[b] |  |  |  |
| obsessive | 1.357 (2.530) | 1.824 (3.206) | 0.662 |
| compulsive | 4.143 (4.258) | 3.688 (3.860) | 0.761 |
| **BDI**[c] |  |  |  |
|  | 3.714 (2.920) | 9.176 (12.259) | 0.115 |
| **STAI-X**[d] |  |  |  |
| state | 38.000 (9.397) | 40.941 (9.523) | 0.396 |
| trait | 37.071 (6.956) | 40.176 (10.212) | 0.342 |
| **BIS11**[e] |  |  |  |
| cognitive | 16.429 (3.322) | 16.412 (3.692) | 0.990 |
| motor | 20.357 (3.934) | 19.882 (5.171) | 0.780 |
| non-planning | 24.571 (5.721) | 24.176 (5.637) | 0.848 |

Mean (standard deviation) for continuous variables and count (%) for categorical variables.

[a]SCID: Structured Clinical Interview for DSM-5

[b]Y-BOCS: Yale-Brown Obsessive Compulsive Scale

[c]BDI: Beck's Depression Inventory

[d]STAI-X: State-Trait Anxiety Inventory

[e]BIS 11: Barratt Impulsiveness Scale Version 11

between the sham and the anodal groups. We also measured the perceived side effects of tDCS; we found no differences between groups in terms of itching, skin irritation, skin pain, fatigue, mood disturbance, and visual distortion (p > 0.05 for all; see S1 Table). However, the intensity of perceived tingling was significantly higher in the anodal group than in the sham group (t(91) = -2.12; p = 0.04). In addition, the degrees of headache and difficulty in concentration were significantly higher in the sham group than in the anodal group (headache: t(91) = 2.84; p = 0.006, concentration: t(91) = 2.15; p = 0.03). The differences in perceived side effects did not affect the behavioral Pavlovian bias. However, there were significant differences in perceived duration and continuity of stimulation between the sham and anodal groups (S2 Table).

## Behavioral results

We used the difference in behavioral accuracy under Pavlovian-congruent and Pavlovian-incongruent conditions to compare the degree of Pavlovian bias across tDCS groups. In the punishment domain, the anodal group did not show any significant difference in behavior under the two punishment conditions (Fig 3A; t(32) = 0.18; p = 0.86). In contrast, the sham group exhibited significantly lower accuracy under the Pavlovian-incongruent punishment condition (e.g., go-to-avoid) than under the Pavlovian-congruent condition (e.g., no-go-to-avoid) (Fig 3B; t(26) = -2.31; p = 0.03). Neither group exhibited a significant difference in accuracy under the two reward conditions (e.g., go-to-win and no-go-to-win). Consistent with these findings, the behavioral Pavlovian bias index in the punishment domain was significantly lower in the anodal group than in the sham group (t(29) = -3.09; p = 0.004).

## Computational modeling

We tested three computational models to explain the data. Model 1 was a reinforcement learning model suggested by Guitart-Masip et al. (2012) which included five parameters ($\xi$: irreducible noise; $\varepsilon$: learning rate; $\rho$: outcome sensitivity; b: go bias; $\pi$: Pavlovian bias). Model 2 was a model with six parameters, including separate feedback sensitivity parameters for reward and punishment cues ($\rho_{rew}$ and $\rho_{pun}$), compared to Model 1. Model 3 further separated Pavlovian bias parameters for reward and punishment cues ($\pi_{rew}$ and $\pi_{pun}$) compared with Model 2. We compared the models using the leave-one-out information criterion (LOOIC) values, which were calculated using leave-one-out cross-validation [47] (Table 2). Data from the sham and

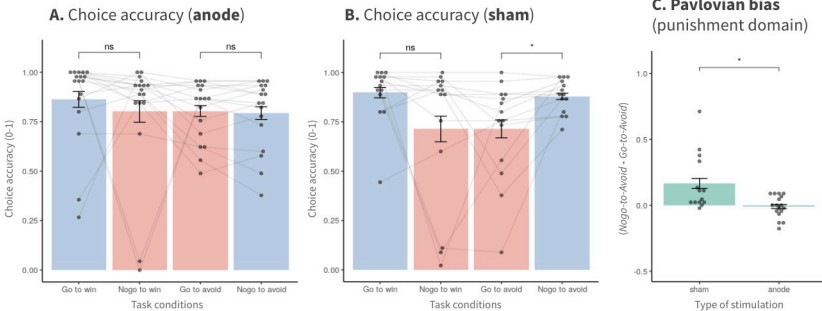

**Fig 3. Pavlovian bias in the punishment domain decreased in the anodal session (accuracy).** In the sham group, we found a significant difference in behavioral accuracy between the punishment conditions, which indicated the presence of Pavlovian bias, particularly in the punishment domain. This difference was not observed in the anodal group. The behavioral index of Pavlovian bias in the punishment domain also showed significantly lower bias in the anodal group than in the sham group. Error bars indicate SEM. *p < 0.05.

**Table 2. Model comparison (LOOIC).**

|  | Parameters | sham | anode |
|---|---|---|---|
| Model 1 | $\xi, \varepsilon, b, \pi, \rho$ | 1852.5 | 2168.7 |
| Model 2 | $\xi, \varepsilon, b, \pi, \rho_{rew}, \rho_{pun}$ | 1813.6 | 2129.7 |
| Model 3 | $\xi, \varepsilon, b, \pi_{rew}, \pi_{pun}, \rho_{rew}, \rho_{pun}$ | **1769.7** | **2093.0** |

Lower LOOIC values indicated better model performance.

anodal groups were fitted separately. Model 3, which had separate Pavlovian bias parameters for reward and punishment domains the best fit (see S2 and S3 Figs in the supplementary material to check one-step ahead prediction results). Models 2 and 1 were the second and third best-fitting models, respectively. Thus, we used estimated parameter values from Model 3 for the subsequent analyses.

## Model parameters

We calculated the posterior distributions of all group-level parameters from Model 3; we compared the results between the anodal and sham groups (Table 3, Fig 4). The anodal group displayed credibly lower irreducible noise ($\xi$), compared with the sham group. Go bias (b) was also credibly lower in the anodal group than in the sham group. Finally, the anodal group had credibly lower Pavlovian bias in the punishment domain ($\pi_{pun}$), compared with the sham group; this is consistent with the behavioral analysis findings that behavioral Pavlovian bias in the punishment domain was significantly lower in the anodal group than in the sham group. Increased involvement of the frontal-striatal network after anodal stimulation might suppress the biases (e.g. Pavlovian bias in the punishment domain, go bias, and choice randomness), thereby interrupting goal-directed behavior of the instrumental system.

However, there were no credible differences between groups in terms of other parameters, such as learning rate ($\varepsilon$), Pavlovian bias in reward domain ($\pi_{rew}$), reward sensitivity ($\rho_{rew}$), and punishment sensitivity ($\rho_{pun}$).

## Discussion

Our results suggest a causal role of the dlPFC in modulating Pavlovian bias in the punishment domain. Moreover, we found that other decision-making tendencies (i.e., go bias and irreducible noise) were also modulated.

**Table 3. Posterior mean (95% HDI highest density interval) of group mean parameters in anodal and sham groups.**

|  | Anode | Sham | Difference |
|---|---|---|---|
| $\xi$<br>irreducible noise | 0.027 [0.009, 0.060] | 0.092 [0.056, 0.141] | -0.064 [-0.117, -0.017] |
| $\varepsilon$<br>learning rate | 0.430 [0.316, 0.554] | 0.321 [0.224, 0.442] | 0.107 [-0.052, 0.264] |
| b<br>Go-bias | -0.173 [-0.689, 0.334] | 1.27 [0.437, 2.46] | -1.45 [-2.72, -0.461] |
| $\pi_{rew}$<br>Pavlovian bias (reward) | 0.045 [-0.223, 0.322] | -0.141 [-0.428, 0.351] | 0.183 [-0.364, 0.587] |
| $\pi_{pun}$<br>Pavlovian bias (punishment) | -0.063 [-0.212, 0.091] | 0.411 [-0.033, 0.855] | -0.473 [-0.939, -0.006] |
| $\rho_{rew}$<br>reward sensitivity | 12.4 [7.72, 21.8] | 12.3 [7.63, 21.4] | 0.146 [-9.91, 10.7] |
| $\rho_{pun}$<br>punishment sensitivity | 6.24 [4.69, 8.31] | 8.83 [5.92, 14.0] | -2.58 [-8.02, 1.02] |

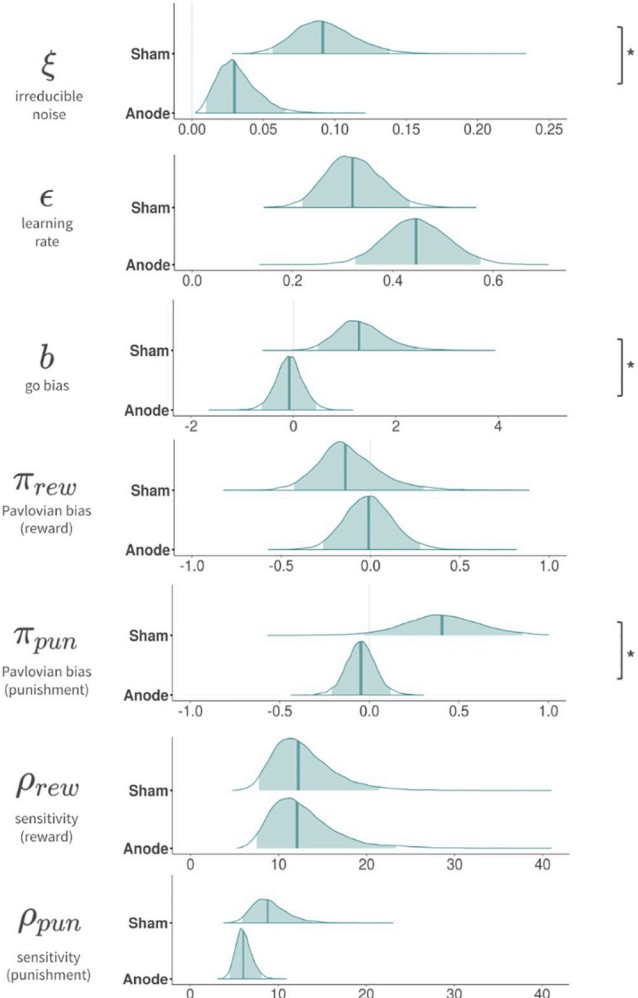

**Fig 4. Pavlovian bias parameter in the punishment domain and other parameters decreased in the anodal session (modeling parameter).** We found lower parameter values in irreducible noise, go bias, and Pavlovian bias in the punishment domain in the anodal session, compared with the sham session. The decrease in Pavlovian bias in the punishment domain is consistent with the behavioral analysis. * 95% highest density interval of the posterior difference did not include zero.

We found that anodal stimulation of the dlPFC reduced Pavlovian bias, which might be related to goal-directed control in the frontal-striatal circuit. The frontal-striatal circuit connects the prefrontal cortex and striatum (including following key areas: ventral striatum [nucleus accumbens], dorsal striatum [caudate and putamen], ventromedial prefrontal cortex, dlPFC, and dorsal anterior cingulate cortex [dACC]) [17, 49]. The dlPFC neurons that project to the striatum may modulate the action-outcome contingency that is encoded and updated in the striatum and ventromedial prefrontal cortex. Therefore, anodal stimulation over the right dlPFC may facilitate high-level cognitive control [4, 5, 12–17] and enhance goal-directed behavior by suppressing Pavlovian bias. Another possible mechanism is that anodal stimulation of the dlPFC increases dopamine release in the striatum [50, 51]; when the dlPFC is stimulated, information about instrumental control is transmitted to the striatum, which responds by increasing dopamine release and overcoming the Pavlovian bias. Furthermore, the tDCS might lead to increased connectivity between the dlPFC and dACC. The dACC, along with the

dlPFC, plays a critical role in updating action values and modulating the integration of subjective value and action-outcome contingency. Previous studies showed that Pavlovian bias was suppressed by frontal midline theta power, an electroencephalography correlate of dACC [7, 8]. Therefore, it is plausible that the tDCS over the dlPFC facilitated the dACC activation, which would reduce Pavlovian bias.

However, the current study only found suppression of Pavlovian bias in the punishment domain, not the reward domain. A recent intracranial study also reported that the electroencephalogram implanted in dlPFC appears to signal punishment learning rather than reward during the instrumental learning task [52]. The anodal stimulation on dlPFC maybe facilitated the value updating for the punishment domain and resulted in a significant change in Pavlovian bias only in the punishment domain. On the other hands, the present findings contribute to knowledge about aversion-related decision-making in the Pavlovian system [53–55]. The underlying neural mechanisms of appetitive-related decision-making have been widely investigated, but the mechanisms that underlie aversive-related decision-making have received less attention [56]. A recent study found that aversive stimuli were associated with active escape response or passive avoidance response [54]. The authors suggested that serotonin might be involved in passive inhibitory responses [57–59], while dopamine might be involved in active escape responses; this is similar to the active approach response toward appetitive stimuli [53, 60]. Therefore, the current results concerning suppression of Pavlovian bias in the aversive domain, obtained by connecting behavioral activation and avoidance (go-to-avoid), might reflect a similar neural process for active escape response. These results are consistent with previous evidence that increased dopamine release after anodal tDCS of the dlPFC might suppress Pavlovian bias [26, 55]. Future tDCS studies should separate avoidance and escape trials to further explore the mechanism that underlies suppression of Pavlovian bias in the punishment domain.

We also observed decreases in go bias and choice randomness in the anodal group. Because go bias and choice randomness interrupt the goal of maximizing benefit, a similar mechanism for interrupting goal-directed behavior may exist, as previously discussed. Increased involvement of the frontal-striatal network (dlPFC, striatum, and dACC) after electrical stimulation of the dlPFC might lead to the suppression of go bias and choice randomness. We presume that the instrumental system may gain preference under conflicting conditions between the instrumental and Pavlovian systems.

Our result showed that decision-making biases were modulated by external intervention, which may have clinical relevance, particularly for substance misuse and other addictive behaviors. For example, increased Pavlovian bias has been linked to substance use and gambling disorders [61, 62], while increased go bias, which may reflect impaired response inhibition, has also been associated with various addictive disorders [63, 64]. Choice randomness (e.g. decision-making noise or inverse temperature) was greater in patients with cocaine abuse and gambling disorders than in healthy individuals [65, 66]. Thus, the current findings may aid in the development of treatments that can reduce the decision-making biases implicated in the various psychiatric conditions. Because the current study only included healthy participants, future studies should include individuals with psychiatric disorders.

One of the potential limitations of the current study is that the second session data were affected by the task practice effect. Therefore, the overall accuracy of task performance was significantly higher in the second session than in the first session. To control for the practice effect, we only analyzed data from the first session and performed between-subject analyses. The resulting smaller sample size might underpower the current findings. For example, while our sham group showed similar behavioral patterns reported in the previous study with healthy controls [9], previously reported Pavlovian bias in the reward domain (go-to-win

versus no-go-to-win) [9] was not replicated, as the bias was trending but not statistically significant. Future studies should attempt to eliminate the practice effect from the experimental protocol and increase the statistical power of the study. Also, we note that the qualitative difference between electric shock and money might make the experimental design and result less consistent with the previous study [9], although electric shock has been used as feedback alongside monetary gain in some studies [34–37]. Lastly, brain regions other than dlPFC were not tested. Contrasting the effects of stimulating other brain regions to stimulating dlPFC might strengthen the causal link between dlPFC and Pavlovian bias. However, it should be also noted that we used a sham-controlled protocol, which is generally considered as effective control condition [33, 67, 68].

In conclusion, our results suggest a causal relationship between non-invasive dlPFC stimulation and corresponding decision-making behavior. We found reduced Pavlovian bias in the punishment domain, go bias, and choice randomness after dlPFC facilitation using anodal stimulation. However, further clarification using neuroimaging techniques is needed to identify the neural mechanism that underlies the effects of tDCS; efforts are also needed to determine how biases are modulated by neural changes in the dlPFC and connected brain networks. In addition, because decision-making biases have been implicated in addictive disorders, our results have practical implications for the treatment of individuals with such disorders. Furthermore, only Pavlovian bias in the punishment, but not the reward domain, was modulated; thus, there is a need for further studies concerning aversive-related decision-making to explain why behavior related to avoiding an aversive state was only modulated by tDCS.

## Supporting information

**S1 Fig. The order effect found in choice accuracy.** When comparing the task performance of the first (third day) and the second time (sixth day) the participants conducted, we found significantly higher accuracy in the second time. This indicates that the second task performance is vulnerable to confounding effects caused by the task order. Therefore, we included only the first task performance to eliminate this potential confounding factor. Error bars indicate SEM. *p < 0.05.
(TIFF)

**S2 Fig. One-step ahead prediction of computational models.** We conducted the one-step-ahead prediction to compare the predicted results and actual choice behaviors. To generate predictions, we utilized 8000 MCMC samples (2000 samples x 4 chains) drawn from individual posterior distributions to predict the probability of a "Go" choice in each of the four conditions, and averaged the predictions with each participant. Model 3 (7-parameter model) emerged as the best performing model, with a mean correlation coefficient of 0.890.
(TIFF)

**S3 Fig. Group prediction accuracy of parameters from Model 2 and Model 3.** Although Model 3 was the best model, Model 2 also showed closely comparable prediction values (mean correlation value = 0.885). The only difference between the two models is the utilization of separate Pavlovian bias parameters for reward and punishment. Consequently, we further compared Models 2 and 3 and investigated whether having two parameters for Pavlovian bias (as in Model 3) would aid in classifying tDCS group membership. By employing ridge logistic regression and leave-one-out cross-validation, we compared the classification accuracy of both models. Model 3 (AUC = 0.806) demonstrated enhanced classification accuracy in comparison to Model 2 (AUC = 0.710). Error bars indicate SEM.
(TIFF)

**S1 Table. Side effect.**
(DOCX)

**S2 Table. Manipulation check of double-blind design.**
(DOCX)

**S1 File. Extended methods.** Includes information on electric shock protocol.
(DOCX)

## Acknowledgments

We thank all the CCS lab members who provided their valuable contributions throughout various stages of this research.

## Author Contributions

**Conceptualization:** Hyeonjin Kim, Jihyun K. Hur, Mina Kwon, Soyeon Kim, Yoonseo Zoh, Woo-Young Ahn.

**Data curation:** Hyeonjin Kim, Jihyun K. Hur, Mina Kwon, Soyeon Kim, Yoonseo Zoh.

**Formal analysis:** Hyeonjin Kim, Jihyun K. Hur.

**Funding acquisition:** Woo-Young Ahn.

**Investigation:** Hyeonjin Kim, Mina Kwon, Woo-Young Ahn.

**Methodology:** Hyeonjin Kim, Woo-Young Ahn.

**Project administration:** Hyeonjin Kim, Woo-Young Ahn.

**Resources:** Woo-Young Ahn.

**Software:** Woo-Young Ahn.

**Supervision:** Woo-Young Ahn.

**Visualization:** Hyeonjin Kim.

**Writing – original draft:** Hyeonjin Kim.

**Writing – review & editing:** Hyeonjin Kim, Jihyun K. Hur, Mina Kwon, Woo-Young Ahn.

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
