## [Decision Letter · Decision Letter 0]

3 Jan 2023

PONE-D-22-23104Causal role of the dorsolateral prefrontal cortex in modulating the balance between Pavlovian and instrumental systems in the punishment domainPLOS ONE

Dear Dr. Kim,

Thank you for submitting your manuscript to PLOS ONE. After careful consideration, we feel that it has merit but does not fully meet PLOS ONE’s publication criteria as it currently stands. Therefore, we invite you to submit a revised version of the manuscript that addresses the points raised during the review process.

We look forward to receiving your revised manuscript.

Kind regards,

Xiong Jiang

Academic Editor

PLOS ONE

Journal Requirements:

This work was supported by the Basic Science Research Program through the National Research  Foundation (NRF) of Korea funded by the Ministry of Science, ICT, & Future Planning (NRF-2018R1C1B3007313 and NRF-2018R1A4A1025891); the National Research Foundation of Korea (NRF) grant funded by the Korea government (MSIT) (No. 2021M3E5D2A0102249311); and the Creative Pioneering Researchers Program through Seoul National University to W.-Y.A.

However, funding information should not appear in the Acknowledgments section or other areas of your manuscript. We will only publish funding information present in the Funding Statement section of the online submission form. 

WYA was supported by the Basic Science Research Program through the National Research Foundation (NRF) of Korea funded by the Ministry of Science, ICT, & Future Planning NRF-2018R1C1B3007313 and NRF-2018R1A4A1025891 (https://www.nrf.re.kr/index); the NRF grant funded by the Korea government (MSIT) No. 2021M3E5D2A0102249311 (https://www.nrf.re.kr/index); and the Creative Pioneering Researchers Program through Seoul National University (https://www.snu.ac.kr/). The funders had no role in study design, data collection and analysis, decision to publish, or preparation of the manuscript.

Reviewers' comments:

Reviewer's Responses to Questions

**Comments to the Author**

1. Is the manuscript technically sound, and do the data support the conclusions?

Reviewer #1: Partly

Reviewer #2: Yes

2. Has the statistical analysis been performed appropriately and rigorously? 

Reviewer #1: Yes

Reviewer #2: Yes

3. Have the authors made all data underlying the findings in their manuscript fully available?

Reviewer #1: Yes

Reviewer #2: Yes

4. Is the manuscript presented in an intelligible fashion and written in standard English?

Reviewer #1: Yes

Reviewer #2: Yes

5. Review Comments to the Author

Reviewer #1: This paper presents a tDCS study of Pavlovian bias using a well-known go/nogo task. The authors show that anodal stimulation of dorsolateral prefrontal cortex disrupts Pavlovian bias for punishment compared to sham stimulation.

Overall, I thought the paper was well-written and the methods were mostly rigorous. I have a few substantive concerns, which I describe below. They render the generality of the study's conclusions somewhat tenuous, but I still think it is worth publishing.

Major comments:

A basic problem with the design used in this study is that because of practice effects the authors are only able to analyze half of their data. This means that the sample sizes are relatively small, making the conclusions less decisive.

Related to the previous concern of small sample size, the authors don't replicate the effect of Pavlovian bias in the appetitive (go-to-win and no-go-to-win) conditions, though the results are trending in that direction. This again makes me think that the study may be underpowered. This should at least be mentioned in the Discussion.

I understand that electric shock is a more powerful aversive stimulus, but I feel that it would have been a clearer comparison between conditions to use monetary gain and loss so that the authors aren't in the position of comparing two qualitatively differently reinforcers. This would have been more consistent with the setup used by Guitart-Masip and colleagues.

The statistics in the results are incompletely reported. In addition to p-values, the authors should report the relevant statistic (e.g., t-statistic, f-statistic), degrees of freedom, etc.

Minor comments:

p. 4: "a previous fMRI studying" -> "a previous fMRI experiment studying"

p. 5: "cortex. ." -> "cortex."

p. 5: "days [23–25]" -> "days [23–25]."

p. 10: 'w' in the numerator of Eq. 1 needs a 't' subscript and should also be capitalized.

p. 10: 'if else' is odd in Eq. 2; better to just say 'otherwise' or 'if a = no-go'.

Reviewer #2: In this manuscript, Kim et al. provides empirical support for a causal role of the dlPFC in reducing the Pavlovian Bias. This is an interesting study, and the paper is well-written.

I have a couple of comments for the authors:

1. The results of figure 3 would be easier to understand if the authors could tell us more about the results of the sham group: are the behavioral results from this group exactly consistent with previous results from the literature in healthy subjects (from Guitard-Massip for instance?). I would recommend to add a paragraph in the discussion to validate this; otherwise, interpreting differences between sham and stimulated groups would be difficult. For example: is it expected to find no Pavlovian bias in the reward condition?

2. The best model has 7 parameters at the end. Did the authors test simpler models and use simulated data against their behavioral data to assess whether beyond statistics metric, it is justified to use such a complex model (for a simple task and behavioral pattern).? Ideally, a simpler model that qualitatively captures the main features of the behavior might be preferred, although this could be discussed (at least simulated data from the competing models could be showed so that this could be evaluated by readers)

3. Related to the previous point, it is not so easy to interpret the computational results because 3 parameters differ between the stimulation groups (sham vs. anode) and it is not clear how these effects interact (the three results are discussed somehow as if this was independent results but it is not…). Could this difficulty be discussed?

4. In the discussion, the authors provide interesting speculation about how their results relate to the neuroscience literature on dlPFC functions; Their results is actually also very consistent with the intracranial study by Gueguen et al (Nature Comm. 2021) in which the electrophysiological response appears to index punishment prediction error signals (more than reward -learning signals) during an instrumental learning task. Could the authors discuss their results in light of these recent evidences?

5. Finally, in my opinion, a (relative) weakness of the experimental design is the lack of a control group for which another brain region would have been stimulated. Perhaps this should be also acknowledged and discussed.

6. PLOS authors have the option to publish the peer review history of their article (what does this mean?). If published, this will include your full peer review and any attached files.

Reviewer #1: No

Reviewer #2: No

---

## [Author Response · Author response to Decision Letter 0]

21 Apr 2023

The more detailed responses to reviewers including figures are written in the 'Response to Reviewers.docx'.

Reviewer #1

C1) A basic problem with the design used in this study is that because of practice effects the authors are only able to analyze half of their data. This means that the sample sizes are relatively small, making the conclusions less decisive. Related to the previous concern of small sample size, the authors don't replicate the effect of Pavlovian bias in the appetitive (go-to-win and no-go-to-win) conditions, though the results are trending in that direction. This again makes me think that the study may be underpowered. This should at least be mentioned in the Discussion.

Response: 

We thank the reviewer for this important comment and suggestion. We agree with the reviewer and added the following sentences and listed it as a limitation of the study:

Page 20: 

“The resulting smaller sample size might underpower the current findings. For example, while our sham group showed similar behavioral patterns reported in the previous study with healthy controls [9], previously reported Pavlovian bias in the reward domain (go-to-win versus no-go-to-win) [9] was not replicated, as the bias was trending but not statistically significant. Future studies should attempt to eliminate the practice effect from the experimental protocol and increase the statistical power of the study.” 

C2) I understand that electric shock is a more powerful aversive stimulus, but I feel that it would have been a clearer comparison between conditions to use monetary gain and loss so that the authors aren't in the position of comparing two qualitatively different reinforcers. This would have been more consistent with the setup used by Guitart-Masip and colleagues.

Response: 

We thank the reviewer for raising this important comment. We used an electric shock as punishment based on previous studies demonstrating that an electric shock produced greater neural activation in regions processing aversive stimuli than monetary loss. We agree with the reviewer that the design led to inconsistency with the Guitart-Masip et al. (2012) study. We revised the following sentences to note the limitation:

Page 8: 

“The feedback included virtual monetary gain as a reward, a yellow bar as a neutral outcome, and an electric shock as punishment instead of monetary loss to maximize the effect of punishment [34].”

Page 21:

“Also, we note that the qualitative difference between electric shock and money might the experimental design and result less consistent with the previous study [9], although electric shock has been used as feedback alongside monetary gain in some studies [34–37].”

References)

Guitart-Masip, M., Huys, Q. J., Fuentemilla, L., Dayan, P., Duzel, E., & Dolan, R. J. (2012). Go and no-go learning in reward and punishment: interactions between affect and effect. Neuroimage, 62(1), 154-166.

C3) The statistics in the results are incompletely reported. In addition to p-values, the authors should report the relevant statistic (e.g., t-statistic, f-statistic), degrees of freedom, etc.

Response:

We thank the reviewer for pointing it out. In the revised manuscript, we reported all relevant statistics and revised sentences are included below.

Page 14:

“However, the intensity of perceived tingling was significantly higher in the anodal group than in the sham group (t(91) = -2.12; p = 0.04). In addition, the degrees of headache and difficulty in concentration were significantly higher in the sham group than in the anodal group (headache: t(91) = 2.84; p = 0.006, concentration: t(91) = 2.15; p = 0.03).”

Page 15:

“In the punishment domain, the anodal group did not show any significant difference in behavior under the two punishment conditions (Fig 3A; t(32) = 0.18; p = 0.86). In contrast, the sham group exhibited significantly lower accuracy under the Pavlovian-incongruent punishment condition (e.g., go-to-avoid) than under the Pavlovian-congruent condition (e.g., no-go-to-avoid) (Fig 3B; t(26) = -2.31; p = 0.03). … Consistent with these findings, the behavioral Pavlovian bias index in the punishment domain was significantly lower in the anodal group than in the sham group (t(29) = -3.09; p = 0.004).”

Reviewer #2

C1) The results of figure 3 would be easier to understand if the authors could tell us more about the results of the sham group: are the behavioral results from this group exactly consistent with previous results from the literature in healthy subjects (from Guitard-Massip for instance?). I would recommend adding a paragraph in the discussion to validate this; otherwise, interpreting differences between sham and stimulated groups would be difficult. For example: is it expected to find no Pavlovian bias in the reward condition?

Response: 

We thank the reviewer for the suggestion. We believe that our results and previous ones (Guitart-Masip et al., 2012) are overall very similar, except that Pavlovian bias in the reward domain (go-to-win versus Nogo to win) is not statistically significant in our study. The difference might be due to a small sample size in our tDCS study and added the following sentences as included below:

Page 20: 

“The resulting smaller sample size might underpower the current findings. For example, while our sham group showed similar behavioral patterns reported in the previous study with healthy controls [9], previously reported Pavlovian bias in the reward domain (go-to-win versus no-go-to-win) [9] was not replicated, as the bias was trending but not statistically significant. Future studies should attempt to eliminate the practice effect from the experimental protocol and increase the statistical power of the study.” 

References)

Guitart-Masip, M., Huys, Q. J., Fuentemilla, L., Dayan, P., Duzel, E., & Dolan, R. J. (2012). Go and no-go learning in reward and punishment: interactions between affect and effect. Neuroimage, 62(1), 154-166.

C2) The best model has 7 parameters at the end. Did the authors test simpler models and use simulated data against their behavioral data to assess whether beyond statistics metric, it is justified to use such a complex model (for a simple task and behavioral pattern).? Ideally, a simpler model that qualitatively captures the main features of the behavior might be preferred, although this could be discussed (at least simulated data from the competing models could be shown so that this could be evaluated by readers)

Response: 

 The reviewer highlighted the importance of evaluating model performance beyond merely relying on statistical metrics. In response, we conducted the one-step-ahead prediction analysis to compare the predicted and actual choice behaviors. To generate predictions for each participant, we utilized 8000 MCMC samples (2000 samples x 4 chains) drawn from individual posterior distributions to predict the probability of a “Go” choice in each of the four conditions, and averaged the predictions within each participant. This approach minimizes sampling bias that could occur when solely using individual posterior means (Haines et al., 2018). Although all three models demonstrated good performance (S2 Fig, attached below), Model 3 (7-parameter model) emerged as the best performing model, with a mean correlation coefficient of 0.890. 

S2 Fig. One-step ahead prediction of computational models. We conducted the one-step-ahead prediction to compare the predicted results and actual choice behaviors. To generate predictions, we utilized 8000 MCMC samples (2000 samples x 4 chains) drawn from individual posterior distributions to predict the probability of a “Go” choice in each of the four conditions, and averaged the predictions with each participant. Model 3 (7-parameter model) emerged as the best performing model, with a mean correlation coefficient of 0.890.

Although Model 3 was the best model, Model 2 also showed closely comparable prediction values (mean correlation value = 0.885). The only difference between the two models is the utilization of separate Pavlovian bias parameters for reward and punishment. Consequently, we further compared Models 2 and 3 and investigated whether having two parameters for Pavlovian bias (as in Model 3) would aid in classifying tDCS group membership. By employing ridge logistic regression and leave-one-out cross-validation, we compared the classification accuracy of both models. Our findings indicate that Model 3 (AUC=0.806) demonstrated enhanced classification accuracy in comparison to Model 2 (AUC=0.710) (S3 Fig, attached below). 

S3 Fig. Group prediction accuracy of parameters from Model 2 and Model 3. Although Model 3 was the best model, Model 2 also showed closely comparable prediction values (mean correlation value = 0.885). The only difference between the two models is the utilization of separate Pavlovian bias parameters for reward and punishment. Consequently, we further compared Models 2 and 3 and investigated whether having two parameters for Pavlovian bias (as in Model 3) would aid in classifying tDCS group membership. By employing ridge logistic regression and leave-one-out cross-validation, we compared the classification accuracy of both models. Model 3 (AUC=0.806) demonstrated enhanced classification accuracy in comparison to Model 2 (AUC=0.710). Error bars indicate SEM.

The results suggest that incorporating separate Pavlovian bias parameters (for reward and punishment) in Model 3 enables a more accurate representation of behavioral patterns within the current dataset, with a significant group difference of Pavlovian bias found only in the punishment domain. 

The figures are included as supplementary figures in the revised manuscript. We also revised the following sentences in the main manuscript as included below.

Page 18: 

“Data from the sham and anodal groups were fitted separately. Model 3, which had separate Pavlovian bias parameters for reward and punishment domains showed the best model fit (see S2 and S3 Figs in the supplementary material to check one-step ahead prediction results).” 

References)

Haines, N., Vassileva, J., & Ahn, W. Y. (2018). The outcome‐representation learning model: A novel reinforcement learning model of the iowa gambling task. Cognitive science, 42(8), 2534-2561.

C3) Related to the previous point, it is not so easy to interpret the computational results because 3 parameters differ between the stimulation groups (sham vs. anode) and it is not clear how these effects interact (the three results are discussed somehow as if this was independent results but it is not…). Could this difficulty be discussed?

Response: 

 We thank the reviewer for pointing out the issue. The noise, go bias, and Pavlovian bias (punishment) parameters are independent constructs but as the reviewer pointed out, they might be related. To investigate this, we examined the correlation coefficients between model parameters (using their individual posterior means), which are plotted in the figure below. Among the three parameters (noise (xi), go bias (b), and Pavlovian bias (piRew & piPun), the highest correlation is 0.5. We also calculated the mean and standard deviation of correlation coefficients across the three parameters, which was 0.46 (0.08). Across all seven parameters, it was 0.08 (0.30).

C4) In the discussion, the authors provide interesting speculation about how their results relate to the neuroscience literature on dlPFC functions; Their results are actually also very consistent with the intracranial study by Gueguen et al (Nature Comm. 2021) in which the electrophysiological response appears to index punishment prediction error signals (more than reward -learning signals) during an instrumental learning task. Could the authors discuss their results in light of these recent evidences?

Response: 

 We thank the reviewer for suggesting the reference and have discussed the reference in Page 19 in the revised manuscript:

Page 19:

“A recent intracranial study also reported that the electroencephalogram implanted in dlPFC appears to signal punishment learning rather than reward during the instrumental learning task [52]. The anodal stimulation on dlPFC maybe facilitated the value updating for the punishment domain and resulted in a significant change in Pavlovian bias only in the punishment domain.”

C5) Finally, in my opinion, a (relative) weakness of the experimental design is the lack of a control group for which another brain region would have been stimulated. Perhaps this should be also acknowledged and discussed.

Response: 

 We thank the reviewer for the comment and suggestion. We revised the manuscript and listed it a limitation:

Page 21:

“Lastly, brain regions other than dlPFC were not tested. Contrasting the effects of stimulating other brain regions to stimulating dlPFC might strengthen the causal link between dlPFC and Pavlovian bias. However, it should be also noted that we used a sham-controlled protocol, which is generally considered as effective control condition [33,67,68].”

---

## [Decision Letter · Decision Letter 1]

22 May 2023

Causal role of the dorsolateral prefrontal cortex in modulating the balance between Pavlovian and instrumental systems in the punishment domain

PONE-D-22-23104R1

Dear Dr. Kim,

We’re pleased to inform you that your manuscript has been judged scientifically suitable for publication and will be formally accepted for publication once it meets all outstanding technical requirements.

Kind regards,

Xiong Jiang

Academic Editor

PLOS ONE

Additional Editor Comments (optional):

Reviewers' comments:

Reviewer's Responses to Questions

**Comments to the Author**

1. If the authors have adequately addressed your comments raised in a previous round of review and you feel that this manuscript is now acceptable for publication, you may indicate that here to bypass the “Comments to the Author” section, enter your conflict of interest statement in the “Confidential to Editor” section, and submit your "Accept" recommendation.

Reviewer #1: All comments have been addressed

2. Is the manuscript technically sound, and do the data support the conclusions?

Reviewer #1: Yes

3. Has the statistical analysis been performed appropriately and rigorously? 

Reviewer #1: Yes

4. Have the authors made all data underlying the findings in their manuscript fully available?

Reviewer #1: Yes

5. Is the manuscript presented in an intelligible fashion and written in standard English?

Reviewer #1: Yes

6. Review Comments to the Author

Reviewer #1: The authors have addressed all of my comments. I just want to call attention to one typo: "money might the"

7. PLOS authors have the option to publish the peer review history of their article (what does this mean?). If published, this will include your full peer review and any attached files.

Reviewer #1: No

---

## [Editor Report · Acceptance letter]

25 May 2023

PONE-D-22-23104R1 

Causal role of the dorsolateral prefrontal cortex in modulating the balance between Pavlovian and instrumental systems in the punishment domain 

Dear Dr. Kim:

I'm pleased to inform you that your manuscript has been deemed suitable for publication in PLOS ONE. Congratulations! Your manuscript is now with our production department. 

Kind regards, 

on behalf of

Dr. Xiong Jiang 

Academic Editor

PLOS ONE